# Detoxified Spent Pot Lining from Aluminum Production as (Alumino-)Silicate Source for Composite Cement and AutoClaved Aerated Concrete

**Arne Peys** [1,*] , **Mateja Košir** [2] , **Ruben Snellings** [1] , **Ana Mladenovič** [2] and **Liesbeth Horckmans** [1]

1   Sustainable Materials Management, VITO, Boeretang 200, 2400 Mol, Belgium; ruben.snellings@vito.be (R.S.); liesbeth.horckmans@vito.be (L.H.)
2   Slovenian National Building and Civil Engineering Institute, Dimičeva 12, 1000 Ljubljana, Slovenia; mateja.kosir@zag.si (M.K.); ana.mladenovic@zag.si (A.M.)
*   Correspondence: arne.peys@vito.be

**Abstract:** New sources of supplementary cementitious materials (SCMs) are needed to meet the future demand. A potential new source of SCM is spent pot lining, a residue from aluminum production. The present work showed that the refined aluminosilicate part of spent pot lining (SPL) has a moderate chemical reactivity in a cementitious system measured in the R3 calorimetry test, comparable to commercially used coal fly ash. The reaction of SPL led to the consumption of $Ca(OH)_2$ in a cement paste beyond 7 days after mixing. At 28 and 90 days a significant contribution to strength development was therefore observed, reaching a relative strength, which is similar to composite cements with coal fly ash. At early age a retardation of the cement hydration is caused by the SPL, which should most likely be associated with the presence of trace amounts of $NH_3$. The spent pot lining is also investigated as silica source for autoclaved aerated concrete blocks. The replacement of quartz by spent pot lining did not show an adverse effect on the strength-density relation of the lightweight blocks up to 50 wt% quartz substitution. Overall, spent pot lining can be used in small replacement volumes (30 wt%) as SCM or as replacement of quartz (50 wt%) in autoclaved aerated concrete blocks.

**Keywords:** supplementary cementitious materials; cement replacement; spent pot lining; autoclaved aerated concrete; cement hydration; lightweight blocks

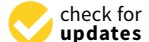

## 1. Introduction

Spent pot lining (SPL) is a hazardous waste originating from the production of aluminum [1,2]. In the Hall–Héroult process, aluminum is reduced electrochemically from alumina using graphite electrodes and a cryolite electrolyte. This process happens in a pot, which has to be replaced every 5 years [1,3]. This results in 2 tonnes of spent pot lining for each 100 tonnes of Al produced adding up to a total of >80,000 tonnes/year in Europe. The lining of the pot is in contact with the cryolite electrolyte. The inner part of the lining is made of graphite and is surrounded by chamotte (aluminosilicate) refractory bricks, the outer part of the lining. During operation the pot lining is slowly impregnated with cryolite and other reaction products forming during aluminum electrolysis. At the end of the lifetime of the pot, the graphite and refractory lining is therefore contaminated with up to 40 wt% of fluorides, around 1 wt% of cyanides and reactive metals and metal oxides that can form explosive gases and toxic leachates in contact with water [1,2,4–6]. The spent pot lining is therefore classified as a hazardous waste (waste classification code 16 11 01*) [7]. Recent developments show that a detoxification process involving a treatment with NaOH can decrease the concentration of contaminants drastically [8,9]. The spent graphite lining can be used in the production of new graphite or in the pyrometallurgical industry. For the application of the chamotte aluminosilicate refractories, several

options exist, however, because of the residual sodium level after the detoxification process, direct application in new refractories is not possible. Efforts have been made to introduce these spent pot lining aluminosilicates in cement production, where it can represent up to 12 wt% of the raw meal [10]. An alternative—not known in literature—is the use of milled material in composite cement as a supplementary cementitious material (SCM). The aluminosilicate composition is desirable for use as SCM. However, further studies are needed to assess whether the refractory part of spent pot lining can contribute to the cement hydration reactions without negative effects on the performance, durability or environmental compatibility.

The embodied greenhouse gas emissions of cement can be lowered by replacing part of the Portland cement clinker with SCMs. The production of Portland cement clinker is associated with 0.8 tonne $CO_2$/tonne clinker [11]. The embodied $CO_2$ emissions of a waste, like spent pot lining, are conventionally seen as zero through the economic allocation set as best practice for life-cycle assessments in ISO 14044. Each wt% of clinker replacement with a waste material thus delivers a reduction of 0.008 ton $CO_2$/ton cement. A replacement level of 30 wt% roughly decreases the $CO_2$ emissions by 30% to 0.56 tonne/tonne cement. On top of this environmental benefit, the durability is often positively impacted by the use of SCMs [12]. In particular aluminosilicate SCMs are known to react with $Ca(OH)_2$ and provide additional hydration products. These additional hydrates fill up space and contribute to the strength development of a mortar or concrete. This space filling is also associated with pore refinement and therefore leads to a lower permeability of the microstructure. A lower permeability is inherently associated with a higher durability in diffusion controlled degradation mechanisms, such as chloride ingress [12,13]. Additionally, the higher Al content of the pore solution when using these aluminosilicate SCMs hinders the dissolution of other silicates, thereby mitigating the alkali-silica reaction. Recent developments show that a synergetic interaction between limestone and aluminosilicate SCMs can lead to an additional decrease in the clinker content of cements [13]. An important characteristic of an SCM to assess whether it can have a beneficial effect on cement hydration is its reactivity [14]. Recently, an efficient test method has been developed in the framework of RILEM TC 267-TRM: the R3 test [15,16]. The SCM is mixed in a solution of $Ca(OH)_2$ with small amounts of $KOH$, $K_2SO_4$ and $CaCO_3$ and the reaction is monitored using isothermal calorimetry or thermogravimetric analysis. The heat release or bound water content is used to assess the reactivity. A good correlation with 28 days strengths has been observed for conventional SCMs.

SCMs rich in silica may be used in the production of autoclaved aerated concrete (AAC) blocks. Autoclaved aerated concrete blocks are porous blocks, which are used as structural insulation material. Commercial products are characterized by a density of 300–800 kg/m$^3$ and compressive strength of 2–12 MPa [17–19]. The density and strength are inversely related to each other. The production process involves mixing cement and/or lime, quartz powder, anhydrite and aluminum powder with water [17,19]. The main components of autoclaved aerated concrete are lime, cement and quartz. The amount of quartz powder in the solid part of the mixture ranges from 50 to 75 wt% [18–20]. Lime and cement can both represent 0–35 wt%. Anhydrite is usually added at 5 wt%, although research exists where blocks are produced without or with partial replacement of the anhydrite [19,20]. The amount of aluminum powder varies between 0.1 and 0.5 wt%, depending on the desired porosity. The water to powder ratio is high: usually water/solid mass ratios of $\geq 1$ are used (although also in this aspect not all literature follows the values for commercial products [19]). These high ratios exceed the amount of liquid needed to obtain a workable mixture, but are needed to enhance the porosity of the autoclaved block. Aluminosilicate residues and byproducts have been investigated to replace the quartz in the mixture by other authors. The addition of aluminates in the system accelerates the reaction kinetics and changes the morphology of the obtained tobermorite phase, while also forming additional hydration products such as katoite [21]. The use of blast furnace slag, coal fly ash and copper tailings have been explored to this end [18,22], suggesting that any

aluminosilicate source may be usable for this application. During precuring the pH rises due to the dissolution of cement and/or lime. This increase in pH accelerates the oxidation of the aluminum powder and reduction of water, resulting in the formation of hydrogen gas bubbles. The mixture expands and a foam is produced. This precured foam is autoclaved to further develop the binder and strength of the block. During curing in the autoclave, usually at 180 °C, the fine quartz reacts with $Ca(OH)_2$ and water and tobermorite is formed, which is the main binding phase that provides strength to the resulting blocks [19,23]. The anhydrite in the starting mixture is needed to promote tobermorite formation [20]. There is no consensus in literature on the mechanism of tobermorite promotion by the addition of anhydrite, but most sources indicate that the influence of sulfates on the morphology and microstructure of the C-S-H intermediate is important [24,25]. The presence of sulfates significantly adapts the pore structure of the binder [26] as it adsorbs to the surface of the C-S-H [27]. On the other hand, the presence of sulfates complicates the recycling of autoclaved aerated concrete blocks, because of environmental leaching of sulfates [28].

This paper focuses on applying the aluminosilicate refractory part of spent pot lining in construction materials. The applications as SCM in composite cement and the use as silica source in AAC blocks are investigated. The reactivity of the aluminosilicate and its influence on cement hydration is determined. The compressive strength evolution of mortar bars is investigated, including the interaction with limestone filler. The use in AAC is explored by comparing the mechanical properties with a benchmark mixture, with focus on the density-compressive strength relation.

## 2. Materials and Methods

### 2.1. Materials

Spent pot lining (SPL) from the Talum aluminum smelter in Kidričevo (Slovenia) was used. The SPL was cut and this study only considers the aluminosilicate part, a.k.a. the second cut. The blocks of SPL were separated among the graphite and aluminosilicate fraction, crushed and grinded to sizes below 4 mm. The treatment procedure of detoxification and extraction of cryolite was carried out in a pilot plant, developed in the frame of the SPL-CYCLE project (http://splcycle.zag.si/, website accessed 19 April 2021). The treatment process in the pilot plant started by filling the extractors with the SPL. The extractors are connected to a continuous water circulation loop. The defined liquid/solid ratio for the aluminosilicate fraction was 1 and the initial treatment step included washing with water and the addition of UV-activated $H_2O_2$ into the circulation loop to decompose the cyanides. The detoxification process was monitored and evaluated with cyanide detection tests (VISOCOLOR ECO; colorimetric test kits). The obtained detoxified leachate was pumped to the collection reservoirs. Afterwards, a fresh batch of water was mixed with NaOH with target concentration of 2 wt% for the extraction of cryolite. Cryolite decomposition was monitored using several electrical conductivity sensors in the liquid circulation loop. The extraction process was completed when the electrical conductivity parameter was stabilized. The obtained leachate was pumped into the collecting reservoir. The aluminosilicate SPL material was washed several times with water using a liquid/solid ratio of 2 until an electrical conductivity close to 500 μS/cm was reached for the water after washing, similar to the initial values of the tap water. The SPL was removed from the extractor and dried at 70 °C overnight. The dried SPL aluminosilicates were milled further to obtain a fineness similar to the Portland cement, aiming for a $d_{50}$ of 10 μm measured using laser diffraction. The laser diffraction measurement was done on a Horiba particle size analyzer, after dispersing the powder in isopropanol and using ultrasound to avoid agglomeration. Ball milling was done for 5 min at 400 rpm in a planetary ball mill to achieve the desired fineness.

### 2.2. Characterization of the SPL

The chemical composition of the SPL aluminosilicates was measured using energy dispersive X-ray fluorescence on a powder sample (ED-XRF). The phase composition

was quantitatively obtained with X-ray diffraction (XRD) using an external standard. An Empyrean diffractometer from PANanalytical was used with a Co target and as parameters during the measurement: 40 kV, 45 mA, 0.0131° 2θ step size and counting time of 0.02 s/step. Qualitative analysis and Rietveld refinement were performed in HighScore X'Pert software. He-pycnometry was done in an AccuPyc II 1340 from Micromeritics to obtain the density of the material. The specific surface area was obtained through BET calculations from a nitrogen sorption experiment on an Autosorb iQ from Quantachrome Instruments. The microstructure of the SPL was studied using scanning electron microscopy (SEM) combined with energy dispersive X-ray spectroscopy (EDS) on a FEI Nova NanoSEM 450. The material was embedded in an epoxy resin, polished and coated with carbon. Images were taken in backscattered electron mode with an acceleration voltage of 20 kV. This voltage setting was also applied during the EDS measurements.

### 2.3. The Influence of SPL on Cement Hydration

The reactivity of the SPL was studied after milling using an R3 heat release test (ASTM C1897-20). In this test 11.11 g of SCM is mixed with 33.33 g of $Ca(OH)_2$, 0.24 g of KOH, 1.2 g of $K_2SO_4$, 5.56 g of $CaCO_3$ and 60 g of demineralized water and the heat release is measured using isothermal calorimetry for 7 days at 40 °C. To study the hydration of mixtures of Portland cement and milled SPL pastes were made. These pastes were composed of 70 g Portland cement (CBR CEM I 52.5R), 30 g of SPL and 40 g of ultrapure water. Reference mixtures with 100 g Portland cement and 70 g Portland cement with 30 g fine quartz filler (Sibelco M400) were studied as well. The hydration kinetics at 20 °C is studied by heat release measurements using isothermal calorimetry on a TAM Air during 28 days. The hydration of the pastes was stopped after 7 and 28 days using the isopropanol-ether solvent exchange procedure recommended for XRD by RILEM TC-238 [29] and studied by XRD and mercury intrusion porosimetry (MIP). MIP was done on a Thermo Fisher Scientific Pascal 240 using a maximum pressure of 200 MPa.

### 2.4. Mixing and Testing of Cement Mortars

The strength was assessed on mortar samples. An optimization was performed using additions of limestone powder ($d_{50}$ = 20 μm). Limestone additions favor the formation of ettringite and monocarboaluminate instead of monosulfoaluminate enabling higher replacement levels [13]. The studied binder mixtures are shown in Table 1, all mortars included 1350 g of CEN standard sand and 225 g ultrapure water. The naming of the samples provides an indication of the wt% of the components (e.g., SPL30-L15-G2 is a mixture with 30 wt% SPL, 15 wt% limestone and 2 wt% gypsum). The mortars were cast as $4 \times 4 \times 16 \text{ cm}^3$ prisms. For each age and mixture, 2 samples were tested for flexural strength and 4 samples in compression at a loading rate of 0.5 mm/min.

**Table 1.** Binder mixtures for strength testing.

| Sample Name | Portland Cement (g) | SPL (g) | Limestone (g) |
|:---:|:---:|:---:|:---:|
| Reference | 450 | | |
| SPL20 | 360 | 90 | |
| SPL30 | 315 | 135 | |
| SPL40 | 270 | 180 | |
| SPL30-L10 | 270 | 135 | 45 |
| SPL30-L15 | 247.5 | 135 | 67.5 |

### 2.5. Mixing and Testing of Aerated Concrete Blocks

AAC blocks were made by casting slurries with binder composition in Table 2 in $4 \times 4 \times 16 \text{ cm}^3$ molds. A water/powder mass ratio of 1 was used. The molds were initially filled to 1.5 cm height. The final height of the sample was dependent on the expansion. The

same Portland cement as for the mortars was used. The lime was high-calcium quicklime from Carmeuse and M400 quartz filler from Sibelco was used for the reference samples. Anhydrite powder was taken from Sigma-Aldrich and aluminum powder from Alfa Aesar ($-325$ mesh, 99.5% purity). Mixing of the dry powder (except aluminum) was performed for 30 s at 300 rpm. The aluminum was premixed with the water by hand. The dry powder and the water-aluminum suspension were mixed for 1 min at 300 rpm. After casting the samples were put without cover in an oven at 40 °C for 4 h. During this precuring step expansion occurred. The precured samples were subsequently placed in an autoclave at 180 °C and 10 bar for 16 h for hardening. The dry bulk density of the hardened specimens was measured after drying at 40 °C by measuring the size of 2 prisms using a caliper and weighing the blocks. The compressive strength was measured on 4 samples. XRD measurements were made of the broken samples from the strength test using the same methodology as described for the raw materials.

**Table 2.** Binder mixtures for the synthesis of autoclaved aerated concrete (AAC) blocks. The complete AAC mixture included 100 g of demi-water.

| Sample Name | Portland Cement (g) | Lime (g) | SPL (g) | Quartz (g) | Anhydrite (g) | Aluminum Powder (g) |
|---|---|---|---|---|---|---|
| Ref 5 | 30 | 15 | | 50 | 5 | 0.1 |
| SPL 25-5 | 30 | 15 | 12.5 | 37.5 | 5 | 0.1 |
| SPL 50-5 | 30 | 15 | 25 | 25 | 5 | 0.1 |
| SPL 75-5 | 30 | 15 | 37.5 | 12.5 | 5 | 0.1 |
| SPL 100-5 | 30 | 15 | 50 | | 5 | 0.1 |
| Ref 3 | 31 | 16 | | 50 | 3 | 0.1 |
| SPL 25-3 | 31 | 16 | 12.5 | 37.5 | 3 | 0.1 |
| SPL 50-3 | 31 | 16 | 25 | 25 | 3 | 0.1 |
| SPL 75-3 | 31 | 16 | 37.5 | 12.5 | 3 | 0.1 |

## 3. Results and Discussion

### 3.1. Characterization of Spent Pot Lining Aluminosilicates

The chemical composition of the SPL was determined using XRF. The results in Table 3 show that the material is almost a pure aluminosilicate, as expected from its refractory origin. The low concentration of sodium may originate from the interaction with the cryolite during operation of the pot or from the NaOH-detoxification process of the spent pot lining. Other minor elements were present as impurities in the original refractory pot lining material.

**Table 3.** Chemical composition of the spent pot lining aluminosilicates determined using XRF.

| | $SiO_2$ | $Al_2O_3$ | $Na_2O$ | $K_2O$ | $Fe_2O_3$ | CaO | MgO | $TiO_2$ |
|---|---|---|---|---|---|---|---|---|
| wt% | 68.0 | 23.8 | 2.8 | 1.8 | 2.4 | 1.4 | 1.3 | 1.1 |

The X-ray diffractogram presented in Figure 1 shows a hump between 20 and 40° 2θ. This is characteristic for an aluminosilicate amorphous phase. The quantification by Rietveld refinement, using an external standard to determine the quantity of amorphous material, is given in Table 4. The high quantity of amorphous (56 wt%) appears promising for the applications, as usually the amorphous fraction of aluminosilicate SCMs such as coal fly ash is the most reactive phase of the material [30]. The major crystalline phases are quartz, mullite and cristobalite. The Na seems to be mostly present in the crystalline phases nepheline and sodalite. Nepheline can also contain minor amounts of potassium [31]. The remaining minor elements could not be associated with a specific phase at this point.

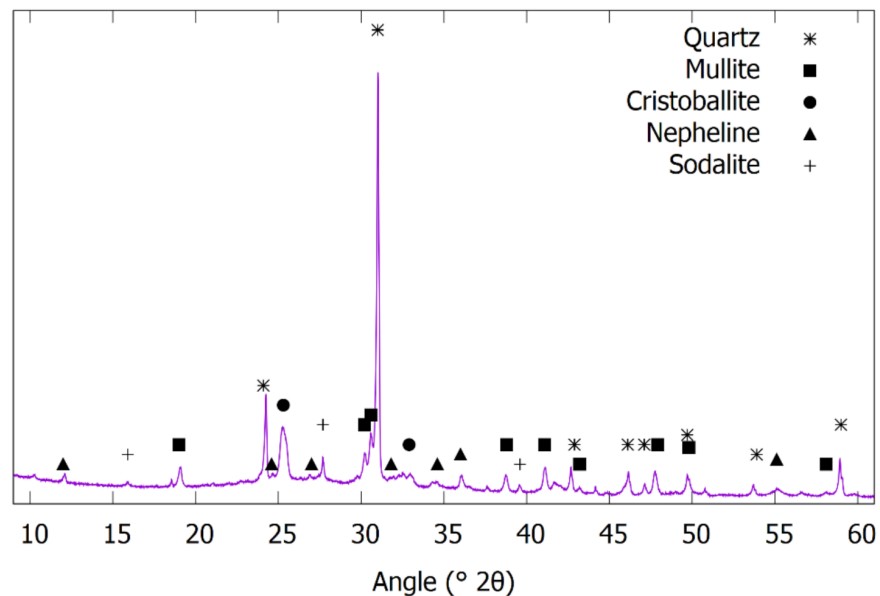

**Figure 1.** X-ray diffractogram of the spent pot lining aluminosilicates.

**Table 4.** Phase composition of the spent pot lining aluminosilicates determined using Rietveld analysis after XRD measurement.

| Phase Name | Phase Formula | Phase Content (wt%) |
|---|---|---|
| Quartz | $SiO_2$ | 17 |
| Mullite | $Al_xSiO_{1.5x+2}$ | 18 |
| Cristobalite | $SiO_2$ | 6 |
| Nepheline | $Na_3KAl_4Si_4O_{16}$ | 2 |
| Sodalite | $Na_8Al_6Si_6O_{25}$ | 1 |
| Amorphous | | 56 |

The distribution of the phases detected by XRD and additional information on the microstructure of the SPL aluminosilicates was obtained using SEM. Backscattered images are shown in Figure 2 and EDS maps are presented in Figure 3. The phases are not homogeneously distributed in the material and most particles are not single-phase particles. The latter is important as this way non-reactive phases can physically hinder the reaction of the amorphous phase. In case of the finer particles, mostly observed on the left image of Figure 2, there might be less intermixing of the crystalline phases with the amorphous phase. The EDS maps underline further the inhomogeneity of the material, especially shown by the localized nature of the Na and Fe maps. The localization of Fe suggests that a small amount of iron oxides might be present, however, this amount was too small to distinguish in the X-ray diffractogram and the presence of iron oxides can thus not be confirmed with certainty.

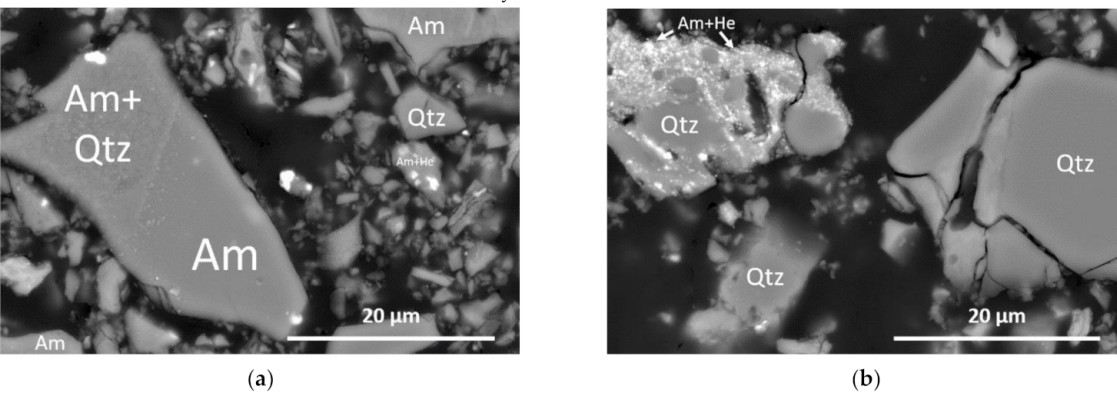

**Figure 2.** Backscattered electron images obtained from electron microscopy on embedded polished powder of the spent pot lining aluminosilicates. Phases that could be recognized are indicated: Am = amorphous, Qtz = quartz, He = hematite. Image (**a**) and (**b**) are two example images from the same sample.

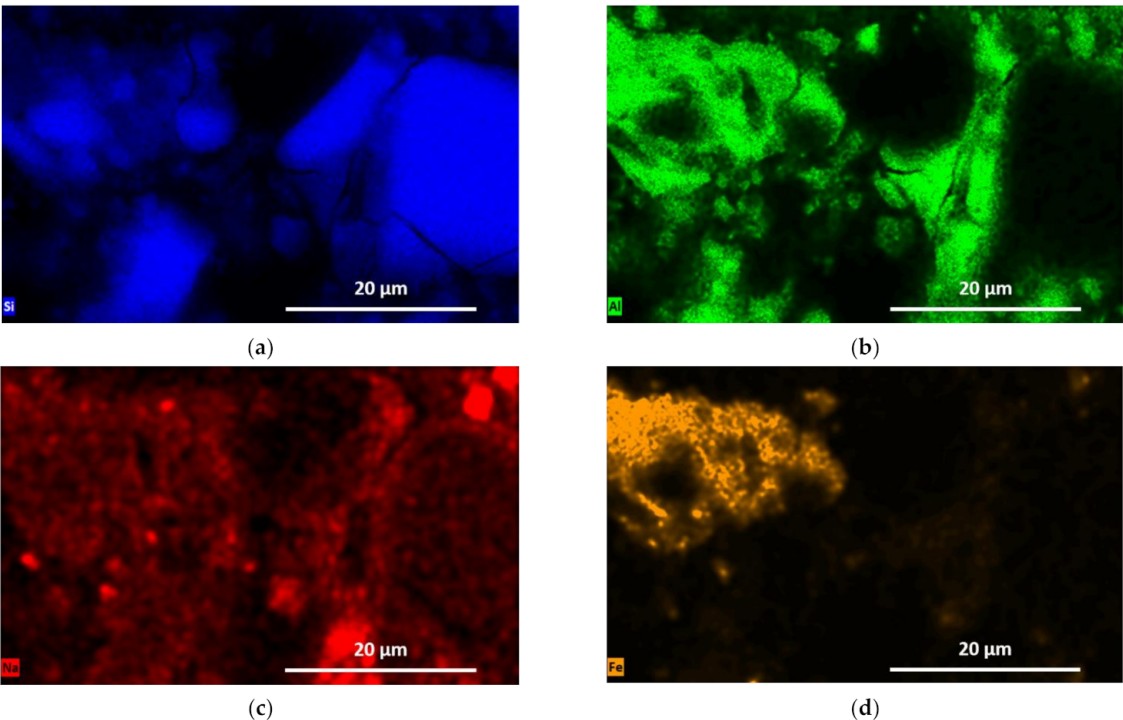

**Figure 3.** Elemental maps of the major elements in spent pot lining aluminosilicates obtained from energy dispersive X-ray spectroscopy measurements: (**a**) Si, (**b**) Al, (**c**) Na and (**d**) Fe. Note that these measurements were performed on image (**b**) in Figure 2.

The physical properties that relate to the performance in the selected application are mainly associated with the fineness of the milled SPL. The SEM images in Figure 2 depict a large fraction of particles below 10 μm. More specifically, the particle size distribution obtained by laser diffraction shows a d50 of 9.8 μm (Table 5). The specific surface of 3.05 m$^2$/g was measured with nitrogen sorption and calculated with BET. The particle size distribution seems slightly coarser than the cement that was used in this study, however, the specific surface of the SPL was higher than that of the cement. The particle shape observed in SEM in Figure 2 helps explain this intuitive discrepancy, as the particles of the milled SPL show relatively high aspect ratios and rough surface textures.

**Table 5.** Physical properties of the milled spent pot lining aluminosilicates. Particle size distribution by laser diffraction, BET specific surface by nitrogen sorption and the true density from He-pycnometry.

| | d10 (µm) | d50 (µm) | d90 (µm) | Specific Surface—BET (m$^2$/g) | Density (kg/m$^3$) |
|---|---|---|---|---|---|
| Milled SPL | 2.3 | 9.8 | 48.0 | 3.05 | 2697 |
| CEM I 52.5R | 1.2 | 7.1 | 26.0 | 1.55 | 3145 |

### 3.2. Spent Pot Lining Reactivity and Influence on Cement Hydration

The reactivity of SPL as SCM was tested using the R3 heat release test at 40 °C. The specific heat presented in Figure 4 evolved towards 200 J/g after 7 days. This is similar to class F coal fly ashes, an SCM commercially used and added in Figure 4 as a reference. This implies that sufficient reactivity is present in the SPL aluminosilicates for use as SCM, provided no other non-cementitious exothermal reactions have occurred. Given the phase composition, the reactivity seems to stem from the aluminosilicate amorphous phase in the SPL.

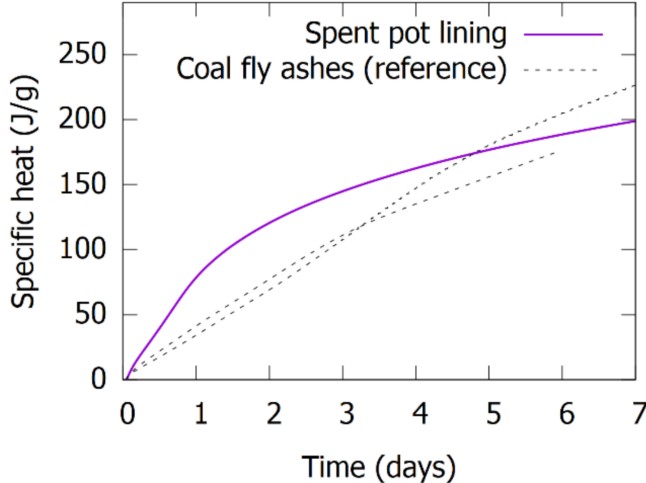

**Figure 4.** R3 heat release of the spent pot lining aluminosilicates determined using isothermal calorimetry compared to class F coal fly ashes as reference. Data for coal fly ashes was obtained from Snellings et al. [32].

The interaction of the SPL aluminosilicates with the hydration of the cement was investigated by studying the reaction kinetics using isothermal calorimetry on cement pastes. The heat flow and heat are both shown in Figure 5, normalized to the amount of CEM I in the mixture. A mixture with 30 wt% of SPL aluminosilicates (70 wt% CEM I, water/binder = 0.4) was compared to two references: one pure CEM I (water/binder = 0.4) and one with 30 wt% quartz filler (70 wt% CEM I, water/binder = 0.4). The sample with quartz filler showed a slight acceleration of the hydration because of the filler effect. This effect also caused a higher cement reaction degree at late age, which can be seen by comparing the total heat at 7–28 days between the CEM I reference and the sample with 30 wt% of quartz filler. The SPL aluminosilicates caused a significant delay in the cement hydration reactions. This is observed most clearly in the heat flow curves, where a shift of the peak is observed. The height and shape of the peak was still quite similar to the references. The shift appeared mainly as a delay of the hydration reaction, when the reactions eventually start, they occur similarly as in the references. Eventually at late age (7–28 days) the cumulative heat was similar to the quartz filler. Whether there was a significant contribution from the reaction of SPL itself is not clear at 28 days, and would require independent measurement of the clinker degree of reaction in the composite cement.

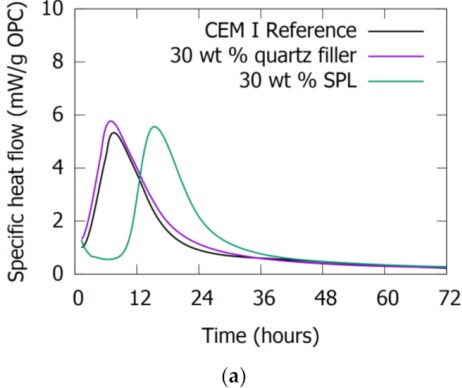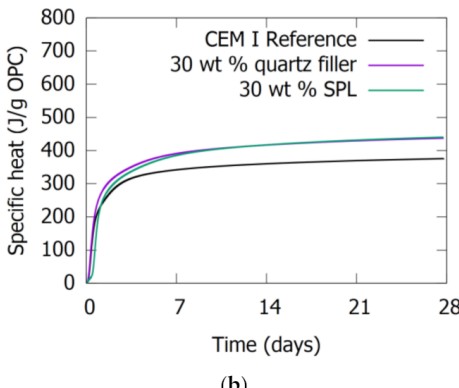

(a)　　　　　　　　　　　　　　　　　　　　　(b)

**Figure 5.** Heat flow (**a**) and cumulative heat (**b**) during isothermal calorimetry measurements. A measurement using 30 wt% spent pot lining aluminosilicates (70 wt% CEM I) is compared to references composed of 100% CEM I and 70 wt% CEM I with 30 wt% fine quartz filler.

The hydration products from the reaction of the cement and SPL were characterized using XRD on samples that were hydration stopped after 7 and 28 days. The diffractograms in Figure 6 show the formation of ettringite and calcium monosulfoaluminate as hydration products. The formation of ettringite before 7 days and afterwards the formation of monosulfoaluminate corresponds to a regular behavior of Portland cement if there are no carbonates added to the mixture [33]. The content of sulfate in ettringite is higher and therefore, this phase formed until the systems ran out of sulfates, after which the formation of monosulfoaluminate started. No qualitative differences with the XRD results of the reference including 30 wt% of quartz filler were observed. The diffractograms of the references and the full diffractograms of the sample with 30 wt% SPL are available in the supplementary information in Figure S1 and S2. The amount of $Ca(OH)_2$ (Portlandite) was investigated in more detail semiquantitatively. A comparison of the height of the peaks in Figure 7 shows that at 7 days a similar amount of $Ca(OH)_2$ was present. At 28 days the amount of $Ca(OH)_2$ in the sample with 30 wt% of SPL was significantly lower than in the sample with 30 wt% quartz filler, indicating the consumption $Ca(OH)_2$. This suggests the pozzolanic reaction of the SPL. The significant $Ca(OH)_2$ consumption was in line with the high heat release observed in the R3 test (Figure 4). The addition of SPL did not result in a change in the type of reaction products, nor was the amount of ettringite or monosulfoaluminate significantly impacted by the use of SPL as SCM, but the pozzolanic reaction of SPL was confirmed by the lower $Ca(OH)_2$ content at 28 days.

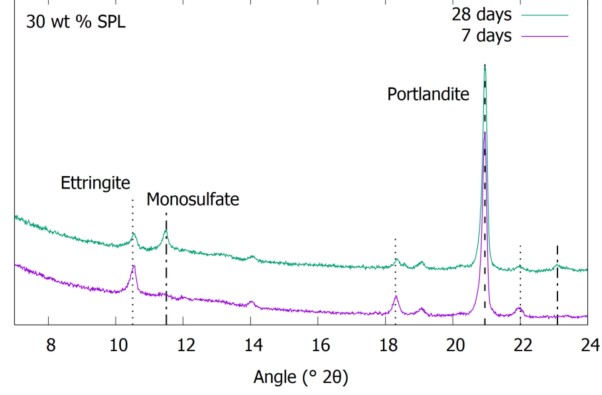

**Figure 6.** X-ray diffractogram of the hydrated pastes including 30 wt% of spent pot lining aluminosilicates. The hydrates are indicated and other peaks result from unreacted SPL.

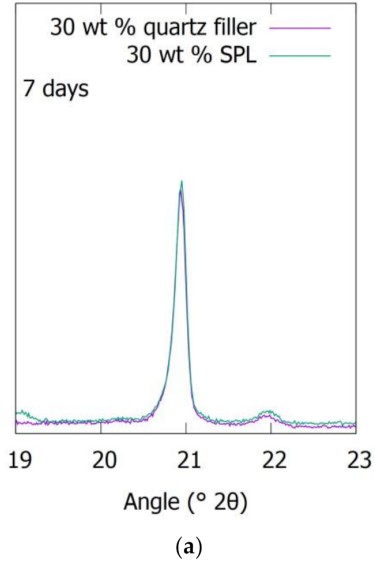 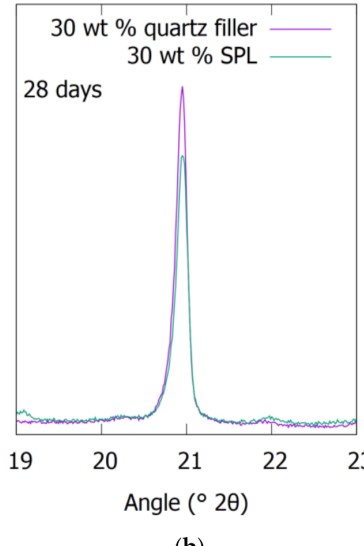

(**a**)        (**b**)

**Figure 7.** Detailed image of the Portlandite peak in the X-ray diffractograms of hydrated pastes including 30 wt% of quartz filler and 30 wt% of spent pot lining aluminosilicates at (**a**) 7 and (**b**) 28 days.

The porosity of the paste samples was studied on 7 and 28 days hydrated cements using MIP. The total porosity decreased by the introduction of 30 wt% SPL as opposed to a quartz filler (Figure 8). The pore volume distribution in Figure 8 shows that the increased porosity is most prominent in the gel porosity below 0.01 μm (10 nm) [34]. The pore filling effect, which is commonly associated to highly reactive SCMs like metakaolin, should be observed as a shift in the threshold pore size—the pore size below which the porosity increases steeply [35]. This effect is observed when comparing 7 and 28 days results, but this is not seen when comparing the pore size distribution of the samples with 30 wt% of SPL and 30 wt% of quartz filler. The reactivity of SPL appears not sufficient to have a significant pore refinement effect on the cement microstructure.

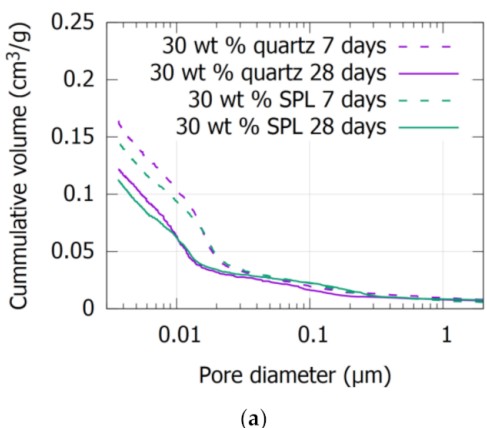 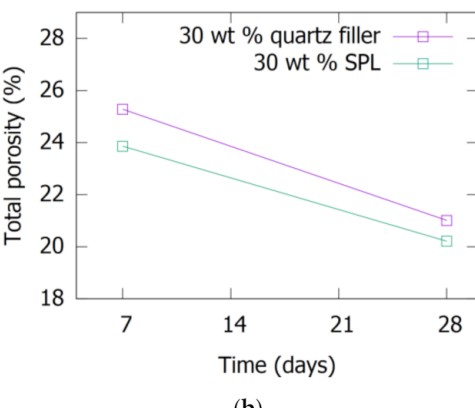

(**a**)        (**b**)

**Figure 8.** Pore volume distribution (**a**) and total porosity (**b**) obtained from mercury intrusion porosimetry measurements. A measurement using 30 wt% SPL 2nd cut (70 wt% CEM I) is compared to a reference composed of 70 wt% CEM I with 30 wt% fine quartz filler.

The SPL aluminosilicates showed moderate chemical reactivity in the R3 test. The pozzolanic behavior of the SPL was confirmed by comparing the amount of $Ca(OH)_2$ using XRD. The pozzolanic reactions did not result in an increase in cumulative heat at 28 days, due to a retarding effect on the heat evolution from Portland cement hydration.

This retardation is mainly observed as a delay of the heat flow peak associated with the hydration reactions. The main components of the SPL aluminosilicates, presented in the characterization in Section 3.1, cannot explain the extensive delay that is observed using calorimetry. A very effective retarder should be present in trace concentrations in the material. The SPL aluminosilicates contain approximately 1 wt% of cyanides, which can have a retarding effect on the hydration of cement [36]. However, as explained in the Materials section, the cyanides were effectively decomposed in the treatment [9], by oxidation using $H_2O_2$. Potential reaction products from this treatment are cyanate $CNO^-$ or ammonium $NH_3$. Furthermore, the possible presence of AlN traces, which are not extractable with the SPL-CYCLE treatment process, could result in their reaction with water and continuous $NH_3$ formation [37]. Ammonium salts are known to delay the hydration of cement and this might thus be a possible explanation for the observed delay [38]. The presence of other potential retarders such as organic acids or sugars is not likely given the origin and treatment of the SPL, while the content of retarding heavy metals such as Zn, Cu or Ni was measured and below levels, which would have a measurable impact [36,39].

### 3.3. Cement Mortars

The compressive strength of mortars was investigated for several replacement levels: 20, 30 and 40 wt%. These are compared to a CEM I reference in Figure 9. A strength drop approximately equal to the replacement level is observed. The addition of limestone (L) slightly improved the contribution of the SPL to compressive strength, as the mixture with 30 wt% SPL and 10 wt% limestone showed higher strength values than the mixture with 40 wt% of SPL. However, the limestone addition was not as effective as observed in the $LC^3$ system [13]. This is expected, as calcined clays show a much higher reactivity than what is observed in this paper for SPL.

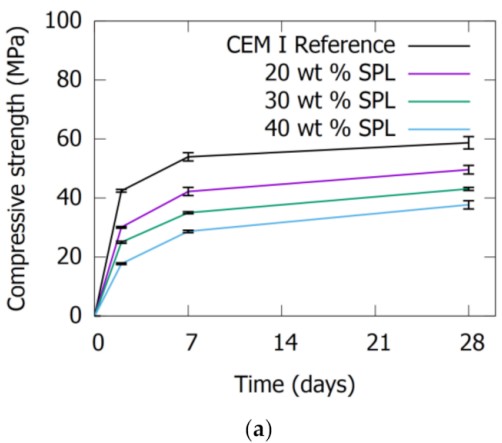
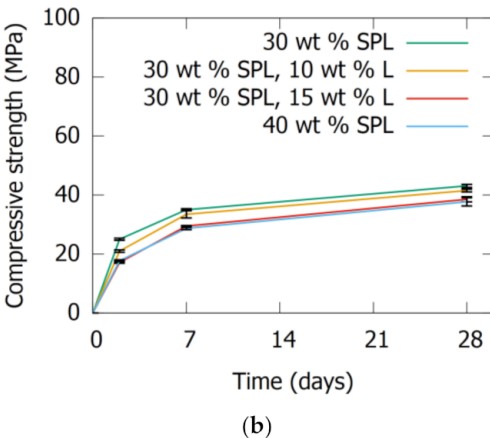

(**a**)　　　　　　　　　　　　　　　　　　　　(**b**)

**Figure 9.** Compressive strength evolution of mortars using several replacement levels of spent pot lining aluminosilicates (SPL) as supplementary cementitious material (**a**). The influence of the addition of limestone (L) to mixtures with 30 wt% of SPL is shown in (**b**).

The strength evolution was studied in more detail for the samples including 30 wt% SPL and including 30 wt% SPL and 10 or 15 wt% limestone in Figure 10. A strength measurement after 90 days was added and the relative strength was calculated (% = strength sample/strength reference sample × 100%). The strength increased significantly after 28 days for the two samples containing SPL and this increase occurred at a higher rate than the reference CEM I. This becomes more clear when observing the calculated relative strengths. The relative strength at an early age (2 and 7 days) lies below the dotted line representing the replacement level, i.e., below 70% strength for 30 wt% SPL and below 55% strength for 30 wt% SPL and 15 wt% limestone. This could be associated with the retarding effect of the SPL on cement hydration, causing a delay in the strength development. The relative strength of the sample including limestone reached its replacement

level at 7 days, showing again the benefit of the limestone addition. At 28 and 90 days, the benefit of the SPL as SCM became clear. The samples with 30 wt% of SPL reached 80% of the strength of the reference at 28 days and 85% of the strength of the reference at 90 days. The continued increase of the relative strength shows a similar behavior to coal fly ashes and other common pozzolanic materials, which are currently used in cement.

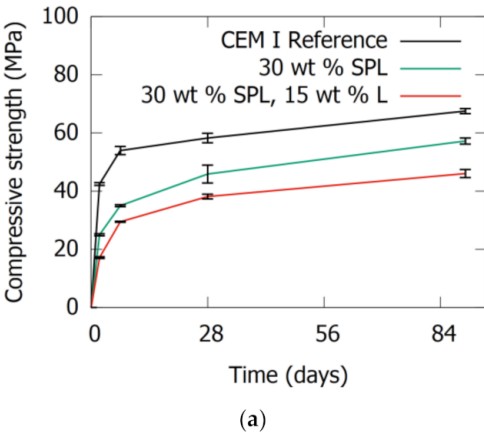 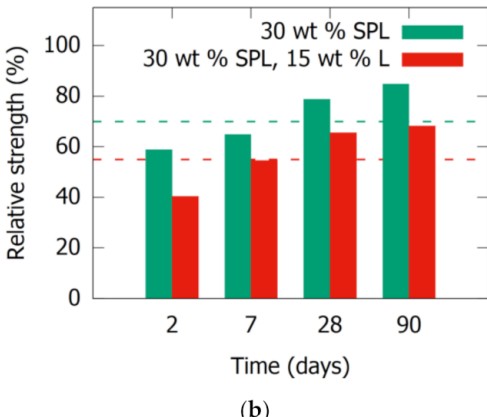

(**a**)          (**b**)

**Figure 10.** Compressive strength evolution of mortars with 30 wt% spent pot lining aluminosilicates (SPL) as supplementary cementitious material, including 90 days strength (**a**). The relative strength is calculated and compared to the replacement level (**b**). The replacement level is represented by a dotted line as a visual aid (in green for 30 wt% replacement—70% strength and in red for 30 + 15 wt% replacement—55% strength).

### 3.4. Autoclaved Aerated Concrete

Autoclaved aerated concrete mixtures were made with 25, 50, 75 and 100 wt% replacement of quartz for SPL. Foaming of all aerated concrete mixtures occurred gradually during the 4 h precuring stage at 40 °C. The slow and steady foaming reaction caused by oxidation of the Al powder and reduction of the water in the paste mixtures of Table 2 is beneficial for the robustness of the industrial process. The foaming reaction resulted in a relatively constant density after autoclaving with most samples showing values around 600 kg/m$^3$, shown in Figure 11. The slight variations in density were mainly caused by the accuracy of weighing the Al powder, as the quantity of Al was the main parameter controlling the porosity next to the water/solids mass ratio, which is more easy to keep accurately constant [19]. The density of all samples falls within the categories defined in EN771-4 for AAC masonry blocks [40]. The slight variations in density, especially the deviation in the samples with 25 and 50 wt% substitution with SPL aluminosilicates and 5 wt% of anhydrite, had to be taken into account when analyzing the strength results.

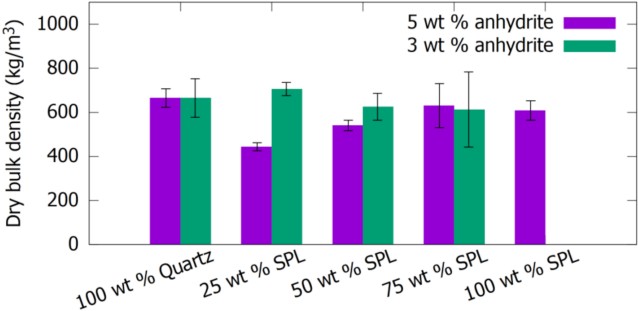

**Figure 11.** Density of the AAC blocks using several replacement levels of quartz for SPL aluminosilicates after drying at 40 °C.

The compressive strength of the blocks is presented in Figure 12. The strength-density relation is plotted on Figure 13. The strength of the reference mixture (100 wt% quartz, 5 wt% anhydrite) reached approximately 5 MPa, which is a decent strength considering the density [18–20,41] and reaches the required strength (4 MPa) for a block of medium density (600–660 kg/m$^3$) according to EN771-4 [40]. The strengths and densities are more easily compared with literature and standard values in Figure 13. This strength was not decreased significantly when using only 3 wt% of anhydrite, which would decrease the sulfate leaching when recycling the block at the end-of-life of the building [28]. The effect of the substitution of the quartz for SPL aluminosilicates is most easily seen for the 3 wt% anhydrite samples, as a similar density was obtained for these samples (Figure 11). The strength did not decrease significantly up to 50 wt% of substitution. The compressive strength still reached 4 MPa for a density of 600 kg/m$^3$. The use of 75 wt% of quartz-SPL substitution resulted in a significant decrease in strength when using 3 wt% of anhydrite. At 5 wt% of anhydrite the compressive strength for the 75 wt% substitution samples was slightly improved, however, also for these samples the strength density relation is not satisfactory and did not reach the values of a commercial AAC strength class according to the EN standard (see Figure 13).

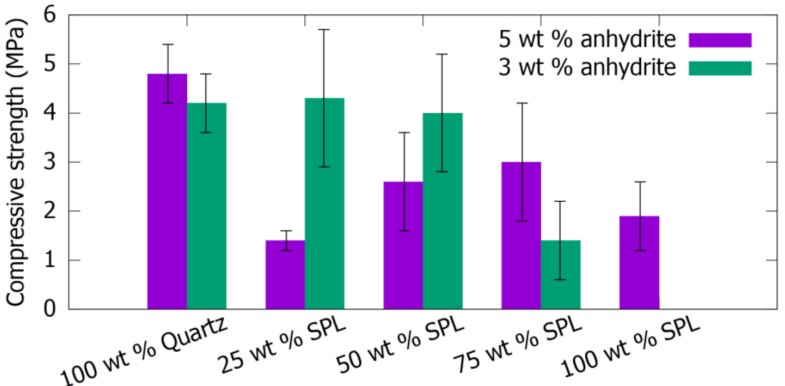

**Figure 12.** Compressive strength of the AAC blocks using several replacement levels of quartz for SPL aluminosilicates.

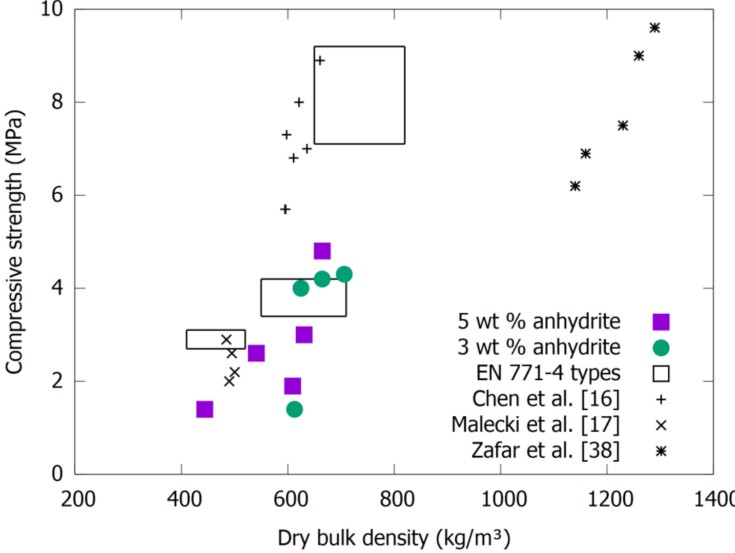

**Figure 13.** Strength-density relation of the AAC blocks compared to the density classes defined in EN 771-4 and to data from references.

The results show that a replacement of quartz for SPL aluminosilicates of 50 wt% was possible without losing significant technical performance in terms of the strength-density relation. Furthermore, the anhydrite content of the mixture could be reduced to 3 wt%, which would enhance the recyclability of the blocks. The 50 wt% of substitution of quartz led to a total SPL content in the raw powder mixture of 25 wt% (see Table 2). The production of AAC is thus a promising route for the valorization of a significant amount of the available SPL. Further optimization of the processing should be steered towards achieving a consistent density and upscaling of the production of the blocks. Additionally, the use of foam stabilizers should be explored, which are usually part of the industrial AAC mixture and can further improve the properties of the blocks.

## 4. Conclusions

This study evaluated whether spent pot lining (SPL) aluminosilicates could be used as raw material in the construction industry after detoxification and extraction of fluoride salts. Testing as supplementary cementitious material (SCM) for composite cements and as raw material in autoclaved aerated concrete (AAC) showed a potential for the valorization in both products. The detoxified SPL was characterized as a partially amorphous aluminosilicate, which after milling showed a moderate reactivity in the R3 heat release test. A cumulative heat release of 200 J/g was obtained after 7 days, which is similar to class F coal fly ashes. When the SPL was mixed in a Portland cement blend, a delay of cement hydration was observed. By 7 days the cumulative heat of the SPL-cement paste caught up with the inert quartz-cement paste. No changes in the hydration product assemblage were observed, but an increased $Ca(OH)_2$ consumption by SPL was detected by 28 days. Mortars showed a significant decrease in compressive strength at 2 days when SPL was used as SCM, but the strength activity index was seen to increase in time, showing that the SPL contributed additional strength by 28 and 90 days. This suggests that cement blends including SPL are promising to be used in general purpose concrete, but further testing at the concrete level is required to confirm their applicability. The use in AAC blocks resulted in a retention of the density-strength relation up to a 50 wt% substitution of quartz for SPL. The production of blocks with 600 kg/m$^3$ and compressive strength of 4 MPa was achieved for these 50 wt% quartz substituted mixes, fulfilling the strength requirements of the medium density type AAC masonry block according to EN771-4. Overall, the use of SPL aluminosilicates was shown to be promising in both applications and the presented work provides a basis for further optimization and technoeconomic studies targeting specific locations and production plants.

**Supplementary Materials:** The following are available online at https://www.mdpi.com/article/10.3390/app11083715/s1, Figure S1: Comparison of the diffractograms of cement pastes with 30 wt% spent pot lining aluminosilicates and 30 wt% of quartz filler at 7 and 28 days, Figure S2: X-ray diffractograms of the samples with 30 wt% quartz filler and 30 wt% spent pot lining aluminosilicates at 7 and 28 days.

**Author Contributions:** Conceptualization, A.P., M.K., R.S., A.M. and L.H.; Data curation, A.P. and M.K.; Formal analysis, A.P. and R.S.; Funding acquisition, A.M. and L.H.; Investigation, A.P.; Methodology, R.S., A.M. and L.H.; Project administration, M.K. and L.H.; Resources, M.K.; Supervision, M.K., R.S., A.M. and L.H.; Visualization, A.P.; Writing—original draft, A.P. and M.K.; Writing—review & editing, R.S., A.M. and L.H. All authors have read and agreed to the published version of the manuscript.

**Funding:** The research in this paper was carried out in the project "SPL-Cycle: Closing the loop of the Spent Pot-line (SPL) in the Al smelting process", a KIC Added Value Activity funded by the European Institute of Innovation and Technology (EIT) Raw Materials (project number 17141). EIT is a body of the European Union and receives support from the European Union's Horizon 2020 Research and Innovation Programme.

**Institutional Review Board Statement:** Not applicable.

**Informed Consent Statement:** Not applicable.

**Data Availability Statement:** The data presented in this study are available on request from the corresponding author.

**Conflicts of Interest:** The authors declare no conflict of interest.

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
