# Peer review of "Detoxified Spent Pot Lining from Aluminum Production as (Alumino-)Silicate Source for Composite Cement and AutoClaved Aerated Concrete"

_applsci, doi:10.3390/app11083715_

Round 1

Reviewer 1 Report

The present paper is very valuable scientific contribution to knowledge on possibility of spent pot lining (SPL) aluminosilicates to be used as raw material in the construction industry after detoxification and extraction of fluoride salts. The contribution can be published in this form. I would like to propose only minor changes that the current state of knowledge of SPL toxicity could be better explained. SPL is hazardous due to toxicity from fluoride and cyanide compounds that are leachable in water, corrosive - exhibiting high pH due to alkali metals and oxides, reactive with water in a way that produces inflammable, toxic and explosive gases, etc.

Author Response

Dear reviewer,

Thank you for your comments. We have done extensive language editing in the revised manuscript to improve the English language throughout the paper.

The discussion on the SPL toxicity has been expanded in the first paragraph of the introduction to accommodate for your suggestion.

Reviewer 2 Report

Remarks:

1.In the introduction, the phrase is incorrect:

 «During operation the pot lining is slowly  impregnated with cryolite». It is saturated not only with cryolite. It is advisable to list all the possible connections recorded in the SPL.

  1. Table 3. Chemical composition of the spent pot lining aluminosilicates determined using XRF. When adding up, it turns out to be above 100%. How can it be?
  2. The experience of research carried out at the Irkutsk National Research Technical University (Russia, Irkutsk) in the application of the refractory part of the SPL after the removal of fluorides in the production of Portland cement has not been analyzed. At least. In Reference authors can indicate the publications of data of researchers on this topic.
  3. In conclusion, I would like to see the ways of further use of the studied material already in the manufacture of finished products (cement).

Author Response

Dear reviewer,

Thank you for your specific comments. We have amended the text to fit these suggestions as follows:

1) The intention of the sentence was to point out that it is because that the cryolite seeps in the lining and reacts with the lining to form the substances mentioned by you and in our manuscript. We have slightly amended the text to avoid confusing other readers. The new sentence in the text reads as follows (new part in yellow): "During operation the pot lining is slowly impregnated with cryolite, resulting in a saturation with cryolite and its reaction products with the pot lining. At the end of the lifetime of the pot, the graphite and refractory lining is therefore contaminated with up to 40 wt% of fluorides, around 1 wt% of cyanides and reactive metals and metal oxides that can form explosive gases in contact with water."

2) The results were not normalized before presenting them in the manuscript. The deviation from 100% originates from experimental errors associated with the XRF measurements and the overestimation of the oxidation state of certain elements (e.g. Fe was presented as Fe2O3, but might be partially as FeO in the material).

3) We have found the following papers from the University of Irkutsk and cited them in the introduction: https://www.atlantis-press.com/proceedings/isees-19/125914224 https://link.springer.com/article/10.1007/s11015-016-0333-4?shared-article-renderer

4) We have added the following sentence to the conclusion section (new part in yellow): "Mortars showed a significant decrease in compressive strength at 2 days when SPL is used as SCM, but the strength activity index was seen to increase in time, showing that the SPL contributed additional strength by 28 and 90 days. This suggests that cement blends including SPL are promising to be used in general purpose concrete, but further testing at the concrete level is required to confirm their applicability."

Reviewer 3 Report

Dear Autors,
I enjoyed reading this paper and found the results interesting. The introduction is clear and frames the need for this research well. Please supplement the literature review with the works of other researchers on this topic or clearly indicate the novelty of these studies. The methods used in the paper are experimental methods. The methods are appropriate for the work and clearly described. The analysis of the results is sufficient. The authors report the outcomes and provide analysis of the results. There is no separate discussion section, but generally the article has the correct structure. English is generally correct, moderate English changes required. Reference to figures and tables has been done correctly. The figures and tables are clear. There are many editorial errors related to citation. E.g line 174, 182, 203, 210, 214... and so on. Instead of the correct citation is "Error! Reference source not found." In line 141 the font is too large. 
To conclude: the article is interesting and valuable, but requires corrected many editing errors.

Author Response

Dear reviewer,

Thank you for your comments. We have reviewed the writing and did some spelling changes throughout the text.

We have corrected the errors in the reference numbering and the formatting in line 141.